# Static palpation ain't easy: Evaluating palpation precision using a topographical map of the lumbar spine as a reference

Inge Strøh Hvidkær[1], Steen Harsted[1,2], Maliheh Hadizadeh[3], Søren O'Neill[1,4], Gregory Neil Kawchuk[5]*, Casper Nim[1,2,4]

**1** Medical Research Unit, Spine Center of Southern Denmark, University Hospital of Southern Denmark, Odense, Denmark, **2** Department of Sports Science and Clinical Biomechanics, University of Southern Denmark, Odense, Denmark, **3** School of Public Health, University of Alberta, Edmonton, Canada, **4** Department of Regional Health Research, University of Southern Denmark, Odense, Denmark, **5** Faculty of Rehabilitation Medicine, Department of Physical Therapy, University of Alberta, Edmonton, Canada

* gkawchuk@ualberta.ca

**Data Availability Statement:** All relevant data are within the manuscript and its Supporting Information files.

## Abstract

### Introduction

Clinicians commonly use manual therapy to treat low back pain by palpating the spine to identify the spinous processes. This study aims to evaluate the ability of experienced clinicians to consistently locate the spinous processes from S1 to T12 through palpation. The results will be compared to topographical data representing the lumbar lordosis at baseline and four follow-up time points.

### Materials and methods

In a prior prospective randomized trial, experienced clinicians used palpation to locate the lumbar spinous processes (S1—T12) and then digitized these locations in three-dimensional space. The same digitizing equipment was then used to continuously collect three-dimensional position data of a wheel that rolled along the back's surface through a trajectory that connected the previously digitized locations of the spinous processes. This process was repeated at 4 days, 1, 4, and 12 weeks.

The resulting lordosis trajectories were plotted and aligned using the most anterior point in the lordosis to compare the locations of the spinous processes identified in different trials. This way, spinous palpation points could be compared to surface topography over time. Intra- and interrater reliability and agreement were estimated using intraclass correlations of agreement and Bland-Altman limits of agreement.

### Results

Five clinicians palpated a total of 119 participants. The results showed a large degree of variation in precision estimates, with a mean total value of 13 mm (95%CI = 11;15). This precision error was consistent across all time points. The smallest precision error was found at

**Funding:** This study was supported by the Foundation for Chiropractic Research and Post Graduate Education in Denmark, which provided financial support for CN's stay at The University of Alberta. They granted 60.000 DKR. The founder had no involvement in the study design, data processing, outline of the manuscript, or the decision to submit the manuscript for publication. The URL to the funders website: https://rltn.dk/fonde/praksisfondene/fond-til-fremme-af-kiropraktisk-forskning-og-postgraduat-efteruddannelse.

**Competing interests:** The authors have declared that no competing interests exist.

L5, followed by S1 File, after which the error increased superiorly. Intra- and interrater reliability was poor to moderate.

## Conclusions

Comparison of palpation results to a topographic standard representing the lumbar lordosis is a new approach for evaluating palpation. Our results confirm the results of prior studies that find palpation of lumbar spinous processes imprecise, even for experienced clinicians.

## Introduction

Low back pain (LBP) is not only common but also a major problem worldwide, as it is the number one cause of years lived with disability [1,2]. Despite this burden, our understanding of LBP remains limited, as approximately 90% of all cases are characterized by non-specific pain [1,3]. Guidelines for treating non-specific LBP encourage different treatment modalities such as manual therapy (MT), exercise and patient education. These modalities can be employed either individually as standalone treatments or in combination [3,4]. However, when clinicians utilizes MT as a treatment option, they often apply static manual palpation to identify the application site for MT as a specific site is considered clinically advantageous [5–7]. In addition, MT is typically given over a series of sessions rather than a single time point. Thus, clinicians who use MT often attempt to provide treatment to the same application site across multiple time points. As such, there is a theoretical need for precise and reliable manual palpation to ensure that the same location is treated over time.

To obtain this certainty, clinicians are taught to utilize static palpation to locate the same spine structures throughout [8]. Results from palpation studies generally show that palpation of spinous processes is imprecise [9–16].

In this study, we used a new approach that employed the topography of the lumbar spine as the standard by which experienced clinicians identified locations of spinous processes. Along that topography, palpation could be assessed for their precision [17]. This approach distinguishes itself from designs typically used in previous validation studies, which have been dedicated palpation studies that investigated the precision of static palpation without a reference [9–11] or with an ultrasonic reference [14,16,18–20], which is not without challenges. Thus, we aimed to utilize the topography of the lumbar spine to detect the precision of palpating spinous process locations (S1 to T12) as individual vertebral levels and as the total length when identified by experienced MT clinicians at five distinct time points.

## Materials and methods

### Design and setting

Data on palpation were extracted from a previous prospective randomized trial involving patients with non-specific LBP [21]. Participants for this secondary analysis (NCT 02868034) were recruited from the University of Alberta through healthcare-seeking and public advertising between February 1st, 2017 and January 30th, 2019. Data collection used in this analysis took place at baseline (i.e. week 0), after 4 days, week 1, week 4, and week 12. Participants were required to have completed the assessment at week 0 and week 1 to be included in the analysis of this study.

The Institutional Review Board granted ethics approval for the primary data collection at the University of Alberta (IRB number: Pro00067152). Informed and written consent was obtained from all participants. However, we do not have consent to publish individual or identifiable clinician data. The study is reported following the *STrengthening the Reporting of OBservational studies in Epidemiology* guidelines [22].

## Data collection

In the prior data collection, each participant assumed a prone position on an examination table, maintaining consistent arm placement. The lumbar spinous processes (S1—T12) were located through palpation performed by one of five experienced clinicians [21] and marked superficially with a felt-tip pen. The position of these markings was then digitized in three-dimensional space and served as reference points for the original study [21]. The same digitizing equipment was then used to continuously collect three-dimensional position data of a wheel that rolled along the back's surface through a trajectory that connected the previously digitized locations of the spinous processes [17,23–25].

The result was a topographical mapping of the lumbar lordosis. The digitization was completed by one of two research assistants who carefully located the center of the location previously marked by the clinician with a device that measures spinal stiffness [17]). Before each assessment, the participants were provided with clear introductions on remaining still and holding their breath during testing. Multiple tests were conducted prior to data collection to familiarize participants with the protocol. This process is illustrated in Fig 1 and virtually via a hyperlink in S1 File.

## Lordosis trajectory alignment

The three dimensional location of the rolling wheel data was collected as it moved between the previously identified spinous process locations using a customized LabVIEW program

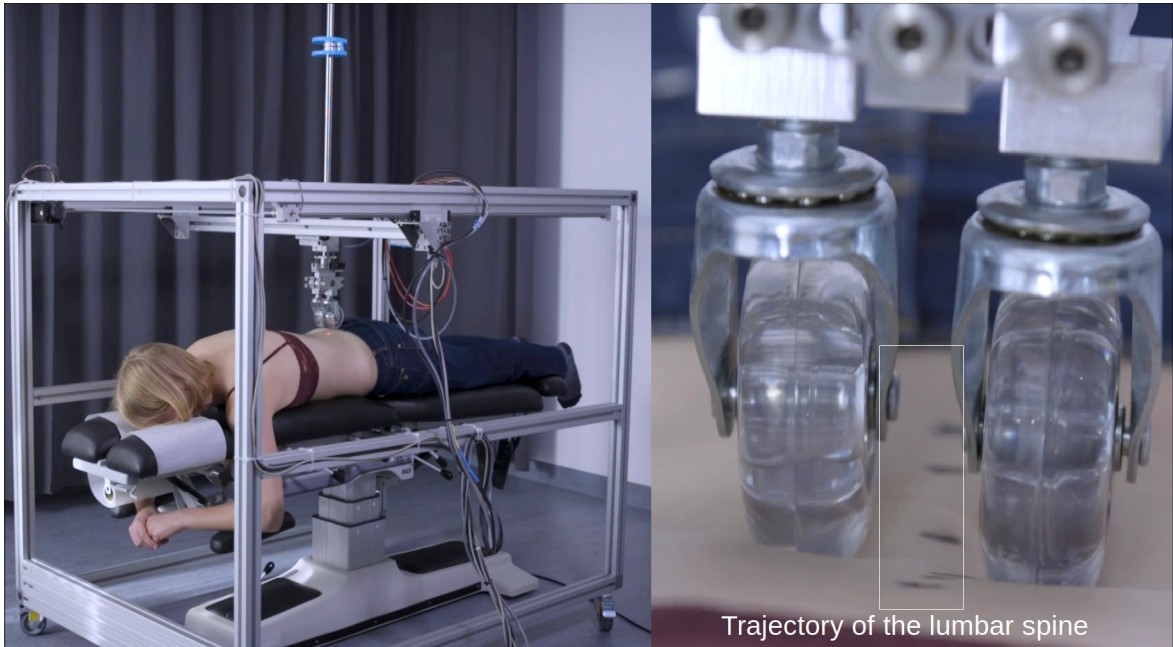

Trajectory of the lumbar spine

**Fig 1. The device that digitized the marked locations of spinous processes.** The device that digitized the marked locations of spinous processes in the lumbar spine (S1—T12) resulting in a three-dimensional lordosis trajectory.

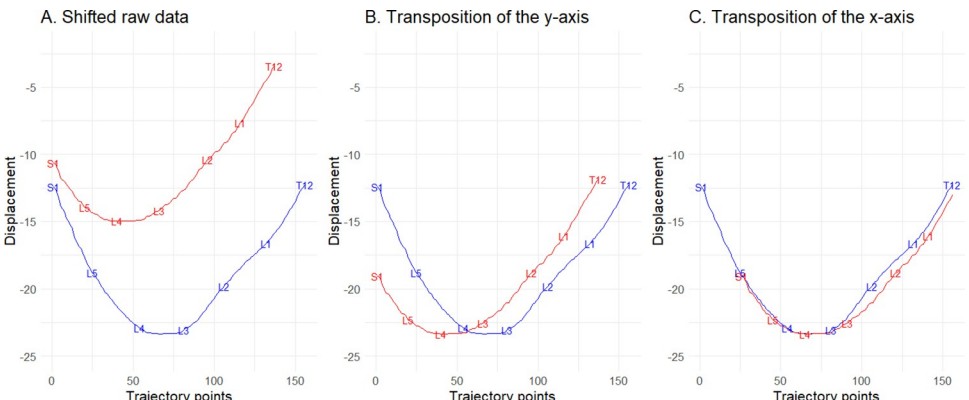

**Fig 2. An overview of the spatial alignment process.** Panel A: Overview of two different trials. Panel B: Overlapping of the y-axis for better visual inspection. Panel C: Aligning the two trajectories by overlapping the x-axis, first done mathematically (corresponding to 25 mm in this case) and then moved manually using visual inspection (if needed).

(version 15, National Instruments, USA). These data were then smoothed using a 2-factor polynomial. Each trial was scrutinized visually to identify and exclude trials that may include participant breathing, movement, voluntary muscle contraction, or technical errors. The formatted raw data were imported into *R* (v. 4.3) [26] and the trajectories were visualized using *ggplot* [27]. Baseline data were used in all cases as the *reference* data (week 0) (Fig 2A). Data collected from subsequent trials were then transposed to align the data points with the reference data. Alignment was achieved mathematically by translating the x-axis until the most anterior points representing the lumbar lordosis were aligned. The magnitude of transposition needed to achieve alignment was then quantified. To better inspect the resulting transposition on the x-axis, we also transposed the y-axis of the plot, so the resulting displacement from the device was equalized (Fig 2B). Finally, based on the visual alignment, two authors (CN, MH) first reviewed the results to inspect whether optimal alignment was achieved. If an agreement was not achieved, the data were further manually transposed and visualized until an agreement was reached. Next, IH went through the data independently of the first two authors, and finally, CN checked all plots again (Fig 2C). Thus, by spatially aligning the lordosis trajectories obtained from each trial, we calculated the average distance between markings of each spinous process (i.e., S1 would always be point 1 on the x-axis at baseline but may have been point -20 at week 1, this would indicate that the clinician was "off" by 20 mm at the week 1 assessment). In addition to each spinous process, we also calculated the total length of the lumbar spine (from S1 to T12). As the multifactorial study's primary concern was unrelated to the precision of palpation, it could be any of the five experienced MT clinicians who completed the palpation on the day of testing.

## Statistical analysis

**Number of participants.** As this study was a secondary analysis using palpation data from a prospective randomized trial [21] no a priori sample size was performed to calculate the number of participants needed to obtain significance, instead we relied on the total patient population at each time point. We did not conduct power analyses, as most of our results in palpation clearly were different from zero.

**Descriptive data.** Participants were described with median (interquartile range) or count (proportion) based on factors potentially related to the difficulty of palpation [28]. These

factors were sex (male/female), age (years), body-mass index (kg/m2), race (collapsed as white, non-white, and multiple races). Additionally, we reported aggregate clinician data.

Next, we assessed the differences in palpation findings compared to the baseline between spinous processes through box plots of each segment for each time point and the total length of the lumbar spine using the same approach.

**Precision of palpation.** We calculated the precision by extracting the values for each vertebra at each time point (i.e., day 4, week 1, week 4, and week 12) with the corresponding vertebra at week 0. We used the absolute difference as the precision, as we were interested in the overall precision and not necessarily whether it was a positive or negative difference from week 0.

To assess the precision of the location of the lumbar spinous processes, we ran linear mixed models with differences in spinous process location as the dependent variable and time interacting with spinous process level as the independent variable and participant as the random error. Results were provided as mean estimates with 95% confidence intervals. We used pairwise deletion for any missing data.

To assess whether participant data were related to precision, we ran univariable linear regression with the summarized mean difference (across time and spinous process level) as the dependent factor and the descriptive data as the independent factor.

All analyses were conducted for precision estimates of each spinous process location and total length of the lumbar spine. No covariates were included for the regression models.

**Clinician variability.** Intra- and interrater reliability and agreement for measurements at L5 and L1 were estimated through comparisons of data collected in week 0 and week 1. The assessment of reliability was conducted using intra-class correlation coefficients of absolute agreement (ICC(2.1)) [29,30]. Agreement was quantified via Bland-Altman limits of agreement (LoA). Additionally, Bland-Altman plots were employed to evaluate the presence of heteroscedasticity [31].

In addition, we also ran univariable linear regression with precision estimates as the dependent variable and the total number of clinicians who provided palpation data and the number of shifts in clinicians (i.e., going from Clinician A to B is one shift, and going from B to A would be a second shift). We used pairwise deletion for any missing data.

**Sensitivity analysis.** We re-ran our *Precision of palpation* analyses as a sensitivity check where we replaced week 0 with week 1 as the reference point. For practical reasons, we used the already aligned data and further shifted it towards the best overlap as determined by a single author (CN).

All analyses were conducted in R (v. 4.3) [26] on a Windows 10 machine using the Tidyverse language [27]. All linear regressions models were checked for assumptions for normality of residuals by inspecting QQ-plots, linearity, and homogeneity of variance. All models upheld these assumptions.

## Results

### Descriptive data

A total of 119 were included in the analysis, 103 (87%) provided data for week 4 and 93 (78%) for week 12. Five clinicians provided palpation data, four were chiropractors, and one was a physiotherapist. All had more than five years of experience and multiple post-graduate courses related to the field of MT. The population mostly consisted of white females in the mid 40's who were slightly overweight (Table 1).

**Table 1. Demographic characteristics of the participants included in the analysis.**

| Participant characteristics | | N = 119 |
|---|---|---|
| Sex [female][1] | | 73 (61%) |
| Race[1] | | |
| | Unknown/more than one race | 20 (17%) |
| | Non-white | 30 (25%) |
| | White | 69 (58%) |
| Age [years][2] | | 43 (32, 53) |
| Body mass index [kg/m²][2] | | 27 (23, 32) |

[1] n (%)

[2] Median (IQR).

## The precision of palpation

There was a large degree of variation in precision estimates, with mean values approximating 13 mm (95%CI = 11 to 15, p-value < 0.01), and this precision error was consistent across all follow-up time points with increases of 0.40 (95% CI = -2.25 to 3.06, p-value = 0.80) at week 1 and increases towards 3.5 (p-value < 0.05) at week 4 and 12. Furthermore, the smallest precision error was found at L5, S1, then L4, and moving cranially until T12, where the largest differences were seen (mean increase of error of 4 mm compared to L5, p-value < 0.01). We also saw a consistent increase in the lumbar spine length over time, reaching a mean increase of 20 mm (p-value <0.01) at week 4 (Fig 3 and Tables 1–4 in S2 File).

Female participants had a tendency towards slightly less precision error compared to men (vertebral level = -2.7 mm [95% CI = -5.7 to 0.23, p-value = 0.07], length = -4.1 mm [-7.6 to -0.55, p-value = 0.02]). None of the other descriptive information was associated with the precision of the palpation (Tables 1–4 in S3 File).

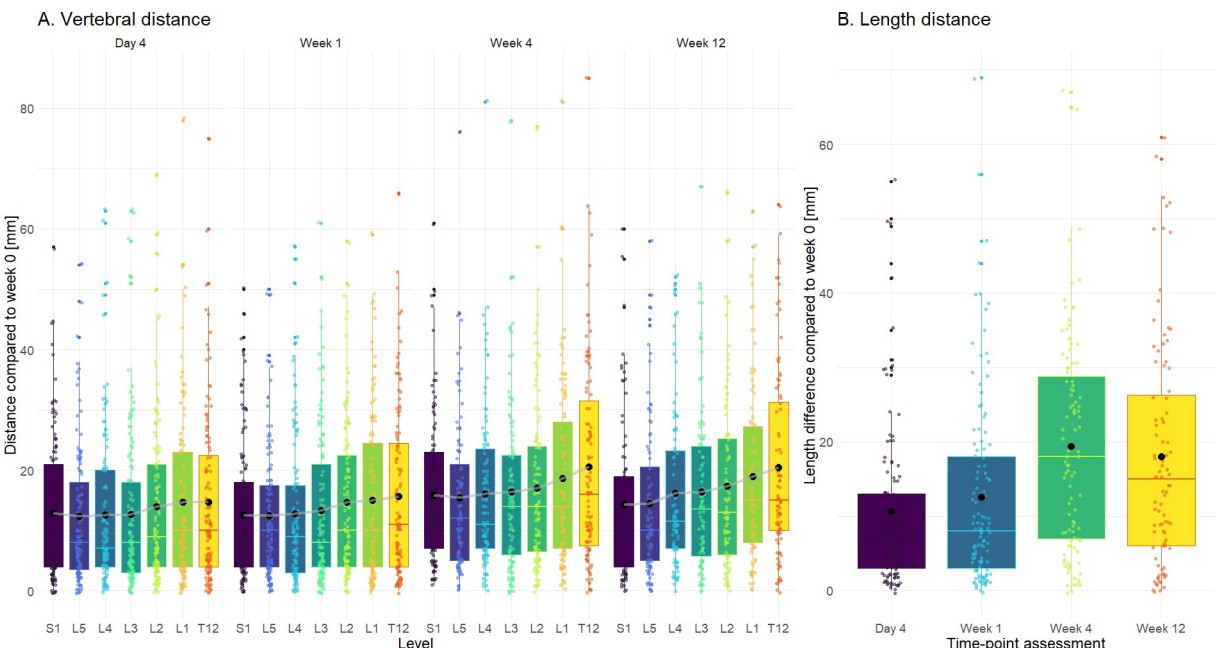

**Fig 3. Distance between each vertebra and lumbar length.** The distance between each vertebra (panel A) and lumbar length (panel B) for each time point compared to week 0 (baseline).

## Clinician variability

A total of 81 and 40 participants underwent measurements at both week 0 and week 1, conducted by the same or different examiners, respectively. Measurements from these participants were included in the analysis of intra- and interrater reliability and agreement. The ICC(2.1) estimates for length measurements taken at L1 by the same examiner were 0.66 (p-value <0.001) and 0.53 (p-value <0.001) if the measurements were taken by different examiners. Both estimates indicate moderate reliability [29]. The Bland-Altman limits of agreement (LoA) for measurements taken at L1 were estimated to range from -35.0mm to 44.6 mm if the measurements were taken by the same examiner and from -39.8mm to 35.6mm if the measurements were taken by different examiners. For length measurements taken at L5, the ICC (2.1) was estimated at 0.18 (p-value = 0.05) in the intrarater analysis and 0.2 (p-value = 0.1) in the interrater analysis. Both estimates indicate poor reliability [29]. The Bland-Altman limits of agreement (LoA) for L5 measurements were initially estimated to range from -31.6 mm to 36.6 mm and from -27.9mm to 30.3mm for the intra- and interrater analysis, respectively. However, upon examination of the Bland-Altman plots, a noticeable trend was observed between the magnitude of the between-session measurements and their differences (see Figs 1–6 in S4 File for detailed plots). To address this trend, LoAs were recalculated post-hoc using the regression approach for nonuniform differences as outlined by Bland & Altman [28]. This adjusted approach yielded similar intrarater LoAs of -1.4mm per unit of the observed value ± 28.4 mm and interrater LoAs of -1.2mm per unit of the observed value ±20.8mm.

The precision estimates were not impacted by the total number of clinicians or the number of shifts (Tables 1 and 2 in S5 File).

## Sensitivity analysis

Applying week 1 as the baseline did not change our results. The mean precision estimate was still at 16.04 mm (95% CI = 13.69, 18.39, p-value <0.01), and no further increases were found in week 12 (Tables 1–6 in S6 File).

## Discussion

We presented a simple method of assessing the precision of palpation by assessing a three dimensional trajectory representing the lordosis of the lumbar spine. We found that there was a substantial amount of variation for the examiners, with mean values of around the length of a single vertebra [32] and the length measured of the lumbar spine increased substantially over time. This imprecision was further underscored by the moderate reliability for L1 measurements and poor reliability for L5.

Interestingly, the Bland-Altman limits of agreement (LoA) were comparable for both L1 and L5, albeit slightly wider for L1. The higher reliability despite similar absolute measurement error suggests that the moderate reliability observed for L1 measurements may be attributed to a larger inter-individual variation in the length measures of L1. The palpation methodology employed by the clinicians in our study reflects current clinical practice and involves initially locating L4/L5 likely by using the anatomical landmark of the tops of the iliac crests [8,33], then S1, and then palpating upwards to identify L1. Given the observed findings, where the measurement error only increased slightly from L5 to L1, a significant portion of the measurement error seems to stem from the initial identification of the starting point, L4/L5, rather than the palpation process up the spine. This suggests that the initial error in locating L4/L5 could propagate upwards, impacting the accuracy of identifying subsequent vertebral levels. Interestingly, this persisted across all participants and clinicians, despite utilizing clinicians

with great experience and who were well-trained on the study procedures. Our results might have shown even more considerable differences if using more novice clinicians [34].

The agreement of intra- and interrater reliability at L1 and L5 is not surprising, as consistency within clinicians' measurements is a prerequisite for agreement across different clinicians. The similar levels of intra- and interrater reliability and agreement suggest that if clinicians struggle to maintain consistency in their own measurements (intrarater reliability), it is logical to expect similar challenges in achieving consensus among different clinicians (interrater reliability).

The validity of this study rests on the rational assumption that observed discrepancies in lordosis trajectories actually reflect mistakes when palpating and marking anatomical landmarks, not a gross anatomical change in lumbar curvature. Arguably, changes could occur due to participant breathing, muscle contraction, or because of the manual treatment itself. Overall, these effects could result in modifications of the lumbar curvature, potentially appearing as non-aligned data (S7 File). However, this does not seem likely, nor appear to be the case, as our sensitivity analysis showed the same results when using week 1 as the reference point.

Previous studies have found both static palpation of the spine and palpation of landmarks to be imprecise [9–16]. The results of our study confirm this, as they show a substantial amount of variation in palpation of the spinous processes of the lumbar spine with a mean difference of around the length of a single vertebra [32] and substantial increases in the length of the trajectory lordosis over time. This study underlines the consequences of inaccurate palpation and unreliable landmark assessment previously published. We add the following new information: the errors are consistent over time and are not limited to missed vertebral locations but also the total length of the lumbar spine. Identifying the starting point of the lumbar spine appears to be a significant cause of this inaccuracy. Surprisingly, whether one or multiple clinicians do the palpation on the same patient appears to be of little value regarding precision.

Newer studies question both the ability and the need to perform MT at a specific application site due to findings that indicate that the pain-reducing effect of, for instance, spinal manual therapy (SMT) is the same both when treatment is applied at a specific "clinically relevant" target and a "not clinically relevant" target–defined by manual palpation [35,36]. Before we can begin to discuss the need to identify specific application sites in treatments, we have to know how well we can actually palpate and find the exact target consistently.

Our results have clinical implications for all professions that rely on static palpation for interventions, such as manual therapists, chiropractors, physiotherapists, etc, but also for other professions such as anesthesiologists who have to make precise indications when providing spinal epidurals [37]. In the field of MT, the effects of being specific to a single vertebral level may not be of particular relevance [35,36] to the treatment effect, so whether a clinician is off by 1 vertebra is likely not important for the effect of the treatment. Nevertheless, it is relevant that the practitioners who perform MT is aware of the low accuracy of static spine palpation as it could affect their understanding of how MT works and what they explain to the patient. However, for anesthesiologists and surgeons ~15mm of imprecision has an impact for their procedures. Therefore, our results emphasized the need for aid in these procedures, (e.g., ultrasound to make a precise injection for spinal epidurals and fluoroscopic placing of needles before spine surgery), which already take place in daily modern practice [38–40].

## Methodological challenges

There are several limitations to consider: First, the spinous processes of the lumbar spine were marked with a pen. Then, the assessor had to register this location using the digitization process. However, all assessors were well-trained in the methodology and attempted to be as

precise as possible. Second, there were multiple clinicians who provided the palpation, and while we were unable to find that shifts between these were associated with less precision, it is unknown if certain clinicians-factors could explain more about lower precision. However, this data was not available due to anonymity issues. Lastly, we aligned the trajectories by manually shifting them based on assessing computerized smoothed plots that were mathematically aligned. We used three raters to conduct the translational process to reduce bias in this approach. Although there may be other ways to complete this spatial synchronization of stiffness data, we have confidence in our approach [17]. Any mistranslations should not have resulted in more than minor changes.

## Conclusion

By comparing palpation results to a topographic standard, we provide a novel insight into the impact that imprecision of static lumbar spinal palpation has and support previous findings while highlighting multiple new aspects surrounding the imprecision of static palpation. Clearly, this is a challenging task associated with substantial variations within and between experienced clinicians and results in not being able to identify the same vertebra consistently and palpating both sacral and thoracic vertebrae as lumbar ones. The starting point appears to be the primary obstacle in precise palpation. Given these results and the limited clinical relevance of static palpation, we suggest that the imprecision of static palpation should be acknowledged in clinical practice where MT is provided and in educational settings.

## Supporting information

**S1 File. A video showing the procedure of measuring the stiffness.**
(DOCX)

**S2 File. The precision of palpation.**
(DOCX)

**S3 File. Association between descriptive information and the precision of the palpation.**
(DOCX)

**S4 File. Intra- and interrater reliability and agreement.**
(DOCX)

**S5 File. Associations with clinician factors.**
(DOCX)

**S6 File. Sensitivity analysis.**
(DOCX)

**S7 File. Data anonymized.**
(CSV)

## Author Contributions

**Conceptualization:** Steen Harsted, Casper Nim.

**Data curation:** Steen Harsted, Casper Nim.

**Formal analysis:** Steen Harsted, Casper Nim.

**Funding acquisition:** Søren O'Neill, Gregory Neil Kawchuk, Casper Nim.

**Investigation:** Inge Strøh Hvidkær, Maliheh Hadizadeh, Casper Nim.

**Methodology:** Inge Strøh Hvidkær, Steen Harsted, Maliheh Hadizadeh, Gregory Neil Kawchuk, Casper Nim.

**Project administration:** Inge Strøh Hvidkær.

**Supervision:** Gregory Neil Kawchuk, Casper Nim.

**Validation:** Casper Nim.

**Visualization:** Steen Harsted.

**Writing – original draft:** Inge Strøh Hvidkær, Casper Nim.

**Writing – review & editing:** Steen Harsted, Søren O'Neill, Gregory Neil Kawchuk.

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
