## [Decision Letter · Decision Letter 0]

8 Mar 2024

PONE-D-24-02050Static palpation ain’t easy: evaluating palpation precision using a topographical map of the lumbar spine as a referencePLOS ONE

Dear Dr. Kawchuk,

Thank you for submitting your manuscript to PLOS ONE. After careful consideration, we feel that it has merit but does not fully meet PLOS ONE’s publication criteria as it currently stands. Therefore, we invite you to submit a revised version of the manuscript that addresses the points raised during the review process.

We look forward to receiving your revised manuscript.

Kind regards,

Barry Kweh

Academic Editor

PLOS ONE

2. In the online submission form you indicate that your data is not available for proprietary reasons and have provided a contact point for accessing this data. Please note that your current contact point is a co-author on this manuscript. According to our Data Policy, the contact point must not be an author on the manuscript and must be an institutional contact, ideally not an individual. Please revise your data statement to a non-author institutional point of contact, such as a data access or ethics committee, and send this to us via return email. Please also include contact information for the third party organization, and please include the full citation of where the data can be found.

3. We note that Figure 1 in your submission contain copyrighted images. All PLOS content is published under the Creative Commons Attribution License (CC BY 4.0), which means that the manuscript, images, and Supporting Information files will be freely available online, and any third party is permitted to access, download, copy, distribute, and use these materials in any way, even commercially, with proper attribution. For more information, see our copyright guidelines: http://journals.plos.org/plosone/s/licenses-and-copyright.

Additional Editor Comments:

An interesting study which compares palpation of the lumbar spine by experienced clinicians with topographical methods. A fundamental flaw of this study that modern surgical practice in most centres would necessitate an XR irrespective of how accurate landmarks have been demonstrated to be useful given every patient is individual. The authors need to justify the utility of their method in everyday clinical practice.

Reviewers' comments:

Reviewer's Responses to Questions

**Comments to the Author**

1. Is the manuscript technically sound, and do the data support the conclusions?

Reviewer #1: Yes

Reviewer #2: Partly

Reviewer #3: Yes

2. Has the statistical analysis been performed appropriately and rigorously? 

Reviewer #1: Yes

Reviewer #2: No

Reviewer #3: Yes

3. Have the authors made all data underlying the findings in their manuscript fully available?

Reviewer #1: No

Reviewer #2: Yes

Reviewer #3: No

4. Is the manuscript presented in an intelligible fashion and written in standard English?

Reviewer #1: Yes

Reviewer #2: Yes

Reviewer #3: Yes

5. Review Comments to the Author

Reviewer #1: Thank you for your efforts in compiling this manuscript. I appreciate the opportunity to review it.

This manuscript presents important findings that demonstrate the low accuracy of static spine palpation by clinicians. To improve the quality of this article, I suggest that you mention the solutions proposed by literature to enhance the accuracy of this method or alternative methods in the discussion section.

Reviewer #2: Hvidkær et al investigate the precision of lumbar spinal palpation by experienced clinicians in treating low back pain. Utilizing data from a previous trial, clinicians marked spinous processes from S1 to T12, and their precision was assessed by comparing these palpation points with a topographical representation of the lumbar lordosis. This is a novel measurement method but does have some concerns about the methodology and clinical meaningful

of results

The 3-d Wheel method although interesting does bring up some concern. It is unclear if this new method of measurement has been validated. If so, this may be added to the method section. The fact that the plots, in addition to computerized smoothing, required manual adjustment in the Y and X plots does introduce the possibility of Bias which may or may not be random.

1. Further uncertainty in measurement is also inherent to this method such as “breathing, movement, voluntary muscle contraction, or technical errors.“, superficial pen marking, and others which the author acknowledges. The fact that the analysis showed trending nonbias errors also adds to this concern.

2. The method does not indicate whether an a priori study was done to estimate # of patients needed to obtain significance, although this may possibly not be possible as they are using data from another already completed study

3. Clarification of what percentage of data was excluded would also be helpful and whether analysis was performed to see if there was bias on the data excluded/incomplete data

4. Although 95% confidence intervals are shown, data does not show p- p-values when comparisons are done, which could be helpful.

5. The wording “Guidelines for treating non-specific LBP often encourage the use of manual therapy (MT) either as a stand-alone treatment or in combination with other non-pharmacological therapies for non-specific LBP (e.g., exercise)” suggests that the main modality is manipulation and exercises as an adjunct. This is debatable and the wording should reflect that exercise is at least as strongly recommended as a main treatment measure vs other measures.

6. The clinical meaningfulness of ~15mm of imprecision in palpation is unclear.

Possible grammatical/formatting errors noted

Line 105: “February 1 st 2017 and January 30th 105 2019” –> “February 1st, 2017 and January 30th, 105 2019 “

Reviewer #3: This manuscript analyzes data generated from a clinical (longitudinal) study to evaluate the ability of multiple clinicians to locate the spinous processes through palpation. The study objectives are important, and the study was approved by the respective IRB/ethics board. The analytical methods proposed look straightforward. My comments are as follows:

(a) Linear mixed models (LMM) were used to fit "difference in spinous process location" as the response variable. LMMs are based on Gaussian assumptions of the random effects and errors. It is not clear if the authors assessed possible non-Gaussian behavior of the random terms, and thus suggest alternatives.

(b) Within the linear mixed model implementation, it is not clear which covariates were controlled for.

(c) In the results section, every claim of significance should be followed by a p-value, thereby providing the strength of significance/non-significance.

6. PLOS authors have the option to publish the peer review history of their article (what does this mean?). If published, this will include your full peer review and any attached files.

Reviewer #1: **Yes: **Sina Afzal

Reviewer #2: No

Reviewer #3: No

---

## [Author Response · Author response to Decision Letter 0]

30 Apr 2024

General comments

We thank the Editor and Reviewers for providing precise and clear review comments. We have provided a point-by-point response and uploaded a tracked-changed manuscript for your perusal. We agree with nearly all of the comments and have updated the manuscript accordingly. In the case where we disagreed, we have attempted to expand on details already within the manuscript to clarify our statements and have provided explicit comments on reasons for not modifying the manuscript. Again, we thank you all for taking the time to review our manuscript with such rigor, and the comments and modifications to the manuscript have definitely improved it. Thank you.

Editor Comments:

Comment E1#0

An interesting study which compares palpation of the lumbar spine by experienced clinicians with topographical methods. A fundamental flaw of this study that modern surgical practice in most centres would necessitate an XR irrespective of how accurate landmarks have been demonstrated to be useful given every patient is individual. The authors need to justify the utility of their method in everyday clinical practice.

Response E1#0

Thank you for the comment. We agree that the results of imprecise static palpation do not have the same consequence on all professions, and where precision is really needed, such as surgical procedures, more rigorous tools are used instead of relying simply on static palpation. In our discussion we have now included additional detail on how the results impact palpation methods in various clinical practices: 

“Our results have clinical implications for all professions that rely on static palpation for interventions, such as manual therapists, chiropractors, physiotherapists, etc, but also for other profession such as anesthesiologists who have to make precise indications when providing spinal epidurals [34]. In the field of MT, the effects of being specific to a single vertebral level may not be of particular relevance [32,33] to the treatment effect, so whether a clinician is off by 1 vertebra is likely not important for the effect of the treatment. Nevertheless, it is relevant that the practitioners who perform MT is aware of the low accuracy of static spine palpation as it could affect their understanding of how MT works and what they explain to the patient. However, for anesthesiologists and surgeons ~15mm of imprecision has an impact for their procedures. Therefore, our results emphasized the need for aid in these procedures, (e.g., ultrasound to make a precise injection for spinal epidurals and fluoroscopic placing of needles before spine surgery), which already take place in daily modern practice [35-37].”

 

Reviewer 1

Comment R1#0

Thank you for your efforts in compiling this manuscript. I appreciate the opportunity to review it. This manuscript presents important findings that demonstrate the low accuracy of static spine palpation by clinicians. To improve the quality of this article, I suggest that you mention the solutions proposed by literature to enhance the accuracy of this method or alternative methods in the discussion section.

Response R1#0

This is a very good point, which we have added at the end of the discussion: 

“Our results have clinical implications for all professions that rely on static palpation for interventions, such as manual therapists, chiropractors, physiotherapists, etc, but also for other profession such as anesthesiologists who have to make precise indications when providing spinal epidurals [34]. In the field of MT, the effects of being specific to a single vertebral level may not be of particular relevance [32,33] to the treatment effect, so whether a clinician is off by 1 vertebra is likely not important for the effect of the treatment. Nevertheless, it is relevant that the practitioners who perform MT is aware of the low accuracy of static spine palpation as it could affect their understanding of how MT works and what they explain to the patient. However, for anesthesiologists and surgeons ~15mm of imprecision has an impact for their procedures. Therefore, our results emphasized the need for aid in these procedures, (e.g., ultrasound to make a precise injection for spinal epidurals and fluoroscopic placing of needles before spine surgery), which already take place in daily modern practice [35-37].”

Furthermore, we have added additional information on how experience impacts our results in the second section of the discussion:

“Interestingly, this persisted across all participants and clinicians, despite utilizing clinicians with great experience and who were well-trained on the study procedures. Our results might have shown even more considerable differences if using more novice clinicians [31].”

Reviewer 2

Comment R2#0

Hvidkær et al investigate the precision of lumbar spinal palpation by experienced clinicians in treating low back pain. Utilizing data from a previous trial, clinicians marked spinous processes from S1 to T12, and their precision was assessed by comparing these palpation points with a topographical representation of the lumbar lordosis. This is a novel measurement method but does have some concerns about the methodology and clinical meaningful of results.

The 3-d Wheel method although interesting does bring up some concern. It is unclear if this new method of measurement has been validated. If so, this may be added to the method section. The fact that the plots, in addition to computerized smoothing, required manual adjustment in the Y and X plots does introduce the possibility of Bias which may or may not be random

Response R2#0

Thank you for this comment, we agree on the concern that you raise. While the 3D method has not been “validated” directly there are multiple validation papers for this device including a bench-top accuracy, and a comfort study. We have cited these and other relevant references. Additionally, a strict protocol was followed during data collection. We also agree that our method where the plots in the Y-axis and X-axis were manually adjustment does add a certain degree of uncertainty in our assessment, which is why we used 3 raters to independently assess the smoothed plots. However, this is a valid concern and we have modified the final statement in the limitations section under the discussion:

"Lastly, we aligned the trajectories by manually shifting them based on assessing computerized smoothed plots that were mathematically aligned. We used three raters to conduct the translational process to reduce bias in this approach. Although there may be other ways to complete this spatial synchronization of stiffness data, we have confidence in our approach [17]. Any mistranslations should not have resulted in more than minor changes."

Comment R2#1

Further uncertainty in measurement is also inherent to this method such as “breathing, movement, voluntary muscle contraction, or technical errors.“, superficial pen marking, and others which the author acknowledges. The fact that the analysis showed trending nonbias errors also adds to this concern.

Response R2#1

We agree with the Reviewer and have already placed emphasizes on these as limitations to our study. Additionally and in relation to the prior comment, we have already expanded on how this might impact our results (please see Response R2#0 above). We have not made additional changes to the manuscript

Comment R2#2

The method does not indicate whether an a priori study was done to estimate # of patients needed to obtain significance, although this may possibly not be possible as they are using data from another already completed study

Response R2#2

As you mention, this study is a secondary analysis of a study that was built on a priori sample size estimates. Therefore, we did not have the possibility to make an a priori study to estimate the needed number of participants to gain significance. We have now elaborated on this in the statistical analysis in the method under the section “Number of participants” 

“As this study was a secondary analysis using palpation data from a prospective randomized trial [21] no a priori sample size was performed to calculate the number of participants needed to obtain significance, instead we relied on the total patient population at each time point. We did not conduct power analyses, as most of our results in palpation clearly were different from zero.”

Comment R2#3

Clarification of what percentage of data was excluded would also be helpful and whether analysis was performed to see if there was bias on the data excluded/incomplete data

Response R2#3

We have inserted percentages with the count values in the descriptive data section. We did not conduct any dropout analyses as we do not consider that dropout would affect our aim. Specific patient characteristics may be related to dropout, however, it should not affect the precision of palpation especially since it was only female characteristics that were associated with lightly less precise palpation. 

Comment R2#4

Although 95% confidence intervals are shown, data does not show p- p-values when comparisons are done, which could be helpful

Response R2#4

Thank you for this comment, we prioritized only showing the 95% confidence intervals as these provide more meaningful estimates of the strength of the association. As our Tables are only placed in the supplementary material, we have inserted the p-values as requested. Additionally, we have also inserted p-values for the ICC models. However, it does not make sense to calculate p-values for limits of agreement 

Comment R2#5

The wording “Guidelines for treating non-specific LBP often encourage the use of manual therapy (MT) either as a stand-alone treatment or in combination with other non-pharmacological therapies for non-specific LBP (e.g., exercise)” suggests that the main modality is manipulation and exercises as an adjunct. This is debatable and the wording should reflect that exercise is at least as strongly recommended as a main treatment measure vs other measures.

Response R2#5

We have modified this sentence in the introduction to make it clear that manual therapy is not preferred over for instance exercise. 

“Guidelines for treating non-specific LBP encourage different treatment modalities such as manual therapy (MT), exercise and patient education. These modalities can be employed either individually as standalone treatments or in combination [3,4]. However, when clinicians utilizes MT as a treatment option, they often apply static manual palpation to identify the application site for MT as a specific site is considered clinically advantageous [(5–7]). 

Comment R2#6

The clinical meaningfulness of ~15mm of imprecision in palpation is unclear.

Response R2#6

Thank you for this comment, we have provided more detailed answer in response E1#0 and R1#0. In the manuscript, we have elaborated on this topic at the end of the discussion:

“Our results have clinical implications for all professions that rely on static palpation for interventions, such as manual therapists, chiropractors, physiotherapists, etc, but also for other profession such as anesthesiologists who have to make precise indications when providing spinal epidurals [34]. In the field of MT, the effects of being specific to a single vertebral level may not be of particular relevance [32,33] to the treatment effect, so whether a clinician is off by 1 vertebra is likely not important for the effect of the treatment. Nevertheless, it is relevant that the practitioners who perform MT is aware of the low accuracy of static spine palpation as it could affect their understanding of how MT works and what they explain to the patient. However, for anesthesiologists and surgeons ~15mm of imprecision has an impact for their procedures. Therefore, our results emphasized the need for aid in these procedures, (e.g., ultrasound to make a precise injection for spinal epidurals and fluoroscopic placing of needles before spine surgery), which already take place in daily modern practice [35-37].”

Comment R2#7

Possible grammatical/formatting errors noted. 

Line 105: “February 1 st 2017 and January 30th 105 2019” –> “February 1st, 2017 and January 30th, 105 2019 “

Response R2#7

Thank you for making us aware. We have rewritten using a Month/Day/Year format, dates now read: February 1st, 2017 etc. 

Reviewer 3

Comment R3#0

This manuscript analyzes data generated from a clinical (longitudinal) study to evaluate the ability of multiple clinicians to locate the spinous processes through palpation. The study objectives are important, and the study was approved by the respective IRB/ethics board. The analytical methods proposed look straightforward. My comments are as follows:

Linear mixed models (LMM) were used to fit "difference in spinous process location" as the response variable. LMMs are based on Gaussian assumptions of the random effects and errors. It is not clear if the authors assessed possible non-Gaussian behavior of the random terms, and thus suggest alternatives.

Response R3#0

Thank you for this comment, we did not initially include this in the manuscript. However, we did check for model assumptions, and this has now been inserted at the very end of the statistical analysis section: “All linear regressions models were checked for assumptions for normality of residuals by inspecting QQ-plots, linearity, and homogeneity of variance. All models upheld these assumptions.” For simplicity, we added the results directly in the method section as well. Thank you for making us aware. 

Comment R3#1

Within the linear mixed model implementation, it is not clear which covariates were controlled for.

Response R3#1

We did not assess any other covariates that were controlled for. We have inserted “No covariates were included for the regression models.” at the bottom of the Precision of palpation section in the method, to clarify this. 

Comment R3#2

In the results section, every claim of significance should be followed by a p-value, thereby providing the strength of significance/non-significance.

Response R3#2

Thank you for this comment, we prioritized only showing the 95% confidence intervals as these provide more meaningful estimates of the strength of the association. However, as our Tables are only placed in the supplementary material, we have inserted the p-values as requested. Additionally, we have also inserted p-values for the ICC models. However, it does not make sense to calculate p-values for limits of agreement

---

## [Editor Report · Decision Letter 1]

15 May 2024

Static palpation ain’t easy: evaluating palpation precision using a topographical map of the lumbar spine as a reference

PONE-D-24-02050R1

Dear Dr. Kawchuk,

We’re pleased to inform you that your manuscript has been judged scientifically suitable for publication and will be formally accepted for publication once it meets all outstanding technical requirements.

Kind regards,

Barry Kweh

Academic Editor

PLOS ONE

Additional Editor Comments (optional):

A well written article which addresses the suggestions proposed by the reviewers. In particular, the usefulness of this static palpation method concerning non-surgical as well as surgical specialties.
---

## [Editor Report · Acceptance letter]

20 May 2024

PONE-D-24-02050R1 

PLOS ONE

Dear Dr. Kawchuk, 

I'm pleased to inform you that your manuscript has been deemed suitable for publication in PLOS ONE. Congratulations! Your manuscript is now being handed over to our production team.

Kind regards, 

on behalf of

Dr. Barry Kweh 

Academic Editor

PLOS ONE